# PCNs, PCBs, and PCDD/Fs in Soil around a Cement Kiln Co-Processing Municipal Wastes in Northwestern China: Levels, Distribution, and Potential Human Health Risks

**DOI:** 10.3390/ijerph191912860

**Published:** 2022-10-07

**Authors:** Jiali Han, Chenyang Xu, Jun Jin, Jicheng Hu

**Affiliations:** 1College of Life and Environmental Science, Minzu University of China, Beijing 100081, China; 2Key Laboratory of Ecology and Environment in Minority Areas (Minzu University of China), National Ethnic Affairs Commission, Beijing 100081, China; 3Beijing Engineering Research Center of Food Environment and Public Health, Minzu University of China, Beijing 100081, China

**Keywords:** persistent organic pollutant, northwestern China, soil, cement kiln, risk assessment

## Abstract

To evaluate the impact of the first cement kiln co-processing municipal wastes in northwest China on the surrounding environment, the concentrations of polychlorinated naphthalenes (PCNs), polychlorinated biphenyls (PCBs), and polychlorinated dibenzo-p-dioxins and dibenzofurans (PCDD/Fs) were determined in 17 soil samples collected around the plant. The concentration ranges of PCNs, PCBs, and PCDD/Fs were 132–1288, 10.8–59.5, and 2.50–5.95 pg/g, and the ranges of toxic equivalents (TEQ) were 1.98–20.8, 2.36–48.0, and 73.2–418 fg/g, respectively. The concentrations of PCNs, PCBs, and PCDD/Fs in this study were generally lower than those in soil around municipal waste incinerators and industrial parks in other areas. An exponential function equation was applied for the relationship between the concentration of the target compounds and the distance from the cement kiln stack, the results showed that PCN and PCB concentrations declined with the increasing of distance from the stack. Furthermore, it was found that the effect of the cement kiln on surrounding soil contaminations with PCNs and PCBs was stronger than that of PCDD/Fs by comparing the PCN, PCB, and PCDD/F homologue profiles in the fly ash sample from the plant and soil samples at different distances. The total carcinogenic risks (CR) of PCNs, PCBs, and PCDD/Fs for children and adults in soil were 1.65 × 10^−8^–8.93 × 10^−8^ and 1.70 × 10^−8^–9.16 × 10^−8^, respectively, which was less than the risk threshold (CR = 1 × 10^−6^), and there was no health risk.

## 1. Introduction

Persistent organic pollutants (POPs) are of global concern due to their persistence in the environment, high toxicity, and biological accumulation [1]. One of the aims of the Stockholm convention, adopted in 2001, is to reduce the impact of POPs on the ecological environment and human health. PCDD/Fs and PCBs are among the first controlled POPs to be listed in the Stockholm convention [2], and PCNs were listed as new POPs in the Stockholm convention in 2015 [3]. For the PCDD/Fs, seventeen 2,3,7,8-substituted PCDD/Fs have been extensively investigated in many countries because of their high toxicity to humans [4]. In addition, some PCNs and PCBs could have toxicity similar to 2,3,7,8-TCDD [5,6]. Therefore, PCNs and PCBs also draw the attention of the public.

PCDD/Fs have never been produced except for for scientific research [7] and are generated unintentionally as by-products of waste incineration, chemical production, metal smelting, and cement production [8,9,10,11]. PCNs and PCBs were used in large quantities as industrial products [12], but have been banned in many countries since the 1970s [13,14]. However, many studies have reported that PCNs and PCBs can also be generated unintentionally during industrial thermal processes, and then may be emitted to the surrounding environment with the flue gas and fly ash, thus posing a health hazard to the surrounding residents [9,15].

Co-processing solid wastes in cement kilns is a method of incinerating solid waste while producing cement. This method can not only achieve the innocent treatment of solid waste, but also provides the heat source for the cement industry and has widespread applications in many countries as a common treatment for municipal waste [16,17,18]. However, the combustion of solid waste inevitably forms POPs such as PCDD/Fs, PCBs, and PCNs, which in turn has an effect on forming and emitting of these compounds during cement production [3,19,20,21]. Due to the persistence, PCNs, PCBs, and PCDD/Fs emitted from cement kilns will cause their successive sedimentation into surrounding soil, and their concentrations in the soil will be continuously accumulated. Therefore, it is necessary to investigate the soil contaminations with PCNs, PCBs, and PCDD/Fs around the cement kiln co-processing municipal wastes, and assess their impact on the surrounding environment.

In this study, we investigated the contamination characteristics of PCNs, PCBs, and PCDD/Fs in soil around the first cement kiln co-processing municipal wastes in Northwest China. First, the concentrations of PCNs, PCBs, and PCDD/Fs in surrounding soil were determined to analyze the trend of their concentrations in soil with distance from the cement kiln. Second, the homologue patterns of PCNs, PCBs, and PCDD/Fs in soils at different distances were compared to analyze the impact of cement kiln co-processing municipal wastes on the surrounding environment. Finally, the potential health risks of residents (adults and children) in this area were assessed by an exposure risk model. Our results will be useful to understand the contamination characteristics of PCNs, PCBs, and PCDD/Fs around cement kiln co-processing municipal wastes.

## 2. Materials and Methods

### 2.1. Sampling

The cement kiln co-processing municipal wastes is located in a valley of the Loess Plateau in northwest China, which is a vulnerable ecological environment area. There are only some residential and farmland areas around the cement kiln without other industrial emission sources. The distance to the adjacent residential and farmland areas is >1 km. The mean annual precipitation is about 511 mm, and the average number of sunny days is 113 per year. The dominant wind direction in this area is west-northwest. However, the plant is located in a valley, and there might be no regular variation of pollutants concentrations with the dominant wind direction. We explain this in the Appendix A.

This plant was established in 2015, using a new dry-process kiln to co-process municipal wastes. The plant mainly uses limestone and clay as raw materials for cement clinkers production, with a 4500 t production capacity per day. The kilns are equipped with an electrostatic precipitator and the dust collected by the dust catcher is finally put back into cement production. The stack is 90 m high, with an online monitoring system. The emission rates of flue gas and dust were approximately 6 × 10^5^ Nm^3^/h and 80 t per year, respectively. During operation, waste is processed at a rate of 110,000 t per year. PCNs, PCBs, and PCDD/Fs could be emitted during the operation, which poses a potential risk to the surrounding residents. The more detailed information of the plant is presented in the Appendix A.

Seventeen soil samples (A1–A17) were collected from the study area in January 2018, and a background soil sample was collected from a nature reserve approximately 15 km away from the plant (Figure 1). In addition, a fly ash sample was collected from the dust catcher at the plant. Using the concentric circles sampling method, 2, 6, 6, and 1 soil samples were collected within 500 m, 500–1000 m, 1000–1500 m, and 1500–2000 m around the plant, respectively, while 2 samples were collected 2000 m away from the plant.

### 2.2. Sample Processing and Instrumental Analysis

The details for methods used to analyze the samples for PCDD/Fs, PCBs, and PCNs in soils are reported in a previous study [22]. Briefly, 20.0 g of freeze-dried soil sample was put into an extraction cell containing a glass fiber filter and added known amounts of internal standards (^13^C_10_-labeled PCNs, ^13^C_12_-labeled PCBs, and ^13^C_12_-labeled PCDD/Fs) (Appendix A), and the samples were extracted by accelerated solvent extraction with hexane/methylene chloride (50:50, *v*/*v*) at 1500 psi and 120 °C. The extraction was concentrated to 1–2 mL by rotary evaporator and then cleaned up by passing it through an acidic silica gel column (from bottom to top: 1.0 g of activated silica gel, 8.0 g of acidic silica gel, 1.0 g of activated silica gel, 4.0 g of anhydrous sodium sulfate) and a multilayer silica gel column (from bottom to top: 1.0 g of activated silica gel, 2.0 g of silica/AgNO_3_ 10% (*w*/*w*) gel, 1.0 g of activated silica gel, 5.0 g of silica gel modified with sodium hydroxide (1 M) (33%, mass fraction), and 1.0 g of activated silica gel, 8.0 g of acidic silica gel, 1.0 g of activated silica gel, 4.0 g of anhydrous sodium sulfate). The target compounds were separated by a basic alumina column (wet column packing). PCBs and PCNs were obtained by 100 mL hexane/methylene chloride (95:5, *v*/*v*) elution, and PCDD/Fs were obtained by 50 mL hexane/methylene chloride (50:50, *v*/*v*) elution. The purified extracts were concentrated to approximately 20 μL using a rotary evaporator and then under a gentle stream of nitrogen gas before being analyzed.

^13^C_10_-labeled PCN, ^13^C_12_-labeled PCB, and ^13^C_12_-labeled PCDD/F internal standards diluted with acetone were added in the fly ash sample (5.0 g), and 1.0 mol/L hydrochloric acid was added dropwise. When hydrochloric acid sufficiently reacted with the fly ash, ultrapure water was used to wash the fly ash until the wash solution pH > 4. The acid washing solution and the water washing solution were collected separately, and then extracted three times with 20 mL dichloromethane after combination. The washed fly ash sample was freeze-dried and then Soxhlet extracted with methylene chloride for 24 h. The extracts were combined and then concentrated to 1–2 mL using a rotary evaporator. The purification methods were the same as soil samples.

The 75 PCNs, 19 PCBs, and 17 2,3,7,8 substituted PCDD/Fs were quantified using a Trace 1310 Gas Chromatography-TSQ 8000 Evo triple quadrupole mass spectrometer (Thermo Fisher Scientific, Waltham, MA, USA). The mass spectrometer was operated using electron ionization and multi reaction monitor mode. The electron energy was 70 eV, and the source temperature was 270 °C. PCN, PCB, and PCDD/F congeners were separated by a DB-5-MS column (60 m × 0.25 mm i. d., 0.25 μm film thickness; Agilent Technologies, Santa Clara, CA, USA) with helium as the carrier gas at 1 mL/min.

### 2.3. Quality Control

The recoveries for ^13^C_10_-labeled PCNs, ^13^C_12_-labeled PCBs, and ^13^C_12_-labeled PCDD/Fs were in the range of 47–104%, 51–82%, and 49–80%, respectively, which were acceptable according to USEPA 1613 and 1668 methods. The limits of quantitation for the PCNs, PCBs, and PCDD/Fs were in the range of 0.03–0.20, 0.24–0.61, and 0.11–0.79 pg/g, respectively. One blank sample was included in each batch of samples, and was treated following the same procedure as used for the soil samples. Some PCNs and PCBs with low molecular weights were detected in the blank sample, but the concentrations of these PCNs and PCBs were all lower than 5% of those in the soil samples. Therefore, the concentrations in soil samples were not corrected with blank values.

### 2.4. Human Health Risk Assessment

The TEQs of PCDD/Fs and PCBs were calculated using the toxic equivalency factors (Appendix A) from the World Health Organization [23]. The toxicities of the PCNs were calculated using the relative response factors of individual PCN congeners to 2,3,7,8-TCDD [24]. There are three major pathways (ingestion, dermal, and inhalation) of human exposure to PCNs, PCDD/Fs, and PCBs associated with soil. The carcinogenic risks (CR) and non-carcinogenic risks (no-CR) of the above three pathways were calculated following the Risk Assessment Guidance for Superfund methodology from the USEPA [25]. The exposure risks of PCNs, PCBs, and PCDD/Fs for residents (children and adults) were assessed by the TEQ concentrations of PCNs, PCBs, and PCDD/Fs [26]. The relevant calculation parameters and equations are presented in the Appendix A.

### 2.5. Statistical Analysis Methods

Principal component analysis (PCA) and correlation analysis (CA) were performed to evaluate the similarity of PCN, PCB, and PCDD/F patterns between fly ash and soil samples. Because the data showed normal distribution, correlation analysis was performed using the pearson correlation analysis in this study. Statistical analyses of the data were performed by SPSS version 23.0 (Armonk, NY, USA).

## 3. Results and Discussion

### 3.1. Concentrations and Congener Profiles of PCNs, PCBs, and PCDD/Fs in the Soil Samples

Table 1 shows the concentrations of PCNs, PCBs, and PCDD/Fs in soil. The total concentrations of 75 PCN congeners (∑_75_PCNs) in soil in this study were in the range of 132–1288 pg/g (mean 409 pg/g) and TEQ concentrations were in the range of 1.98–20.8 fg/g (mean 6.14 fg/g). For PCBs, 12 dioxin-like PCBs (CB-77, CB-81, CB-105, CB-114, CB-118, CB-123, CB-126, CB-156, CB-157, CB-167, CB-169, and CB-189), 7 indicator PCBs (CB-28, CB-52, CB-101, CB-118, CB-138, CB-153, and CB-180) and CB-209 were detected. The total concentrations of the 19 PCBs congeners in soil were in the range of 10.8–59.5 pg/g (mean 30.6 pg/g), and TEQ concentrations were in the range of 2.36–48.0 fg/g (mean 12.0 fg/g). The total concentrations of seventeen 2,3,7,8-PCDD/F congeners in soil were in the range of 2.50–5.95 pg/g (mean 3.84 pg/g) and TEQ concentrations were in the range of 73.2–418 fg/g (mean 177 fg/g). The concentrations of PCNs, PCBs, and PCDD/Fs in the background sample of this study were 40.0 pg/g, 11.0 pg/g, and 1.90 pg/g, respectively. The concentrations of PCNs, PCBs, and PCDD/Fs in the soil samples were generally greater than those in the background sample. PCNs were the most dominant POPs in soil, contributing more than 90% of total concentrations. Similar results have been found in previous studies. Xu et al. detected the concentration of PCNs, PCBs, and PCDD/Fs in an industrial park in Ningxia Province (China), and found that the PCNs were also the dominant contributor of total concentrations, accounting for 87.6% [27]. Wu et al. reported that the concentration of PCNs in soil was much higher than PCDD/Fs and PCBs in an industrial park in Shandong Province (China), accounting for 91.6% [28]. The total concentration of ∑TEQ (PCNs + PCBs + PCDD/Fs) in soil ranged from 80.9 to 437 fg/g (mean 195 fg/g). Although PCNs contributed the most to the total concentrations of POPs, PCDD/Fs were the largest contributor to the total TEQ (Appendix A). The average ΣTEQ concentration in soil was under the soil quality guideline from Canada (4 pg TEQ/g) [29].

The congener patterns of PCNs, PCBs, and PCDD/Fs were similar in most soil samples. CN-2 and CN-5/7 were the dominant congeners, contributing 13.1% and 11.6% to ∑_75_PCNs, respectively. The indicator PCBs (85.4%) contributed higher than dioxin-like PCBs (11.8%) and CB-209 (2.8%). CB-28 (64.9%) was the dominant indicator PCBs contributor, and CB-118 (27.9%), CB-77 (19.2%), and CB-105 (17.6%) were the main dioxin-like PCB congeners. For PCDD/Fs, the main congeners were octa-CDD (OCDD), 1,2,3,4,6,7,8-hepta-CDF (HpCDF), and octa-CDF (OCDF), which contributed 43.2%, 13.2%, and 12.2%, respectively.

The concentrations of PCNs, PCBs, and PCDD/Fs in this study were compared with those reported in previous studies (Table 2). The concentrations of PCNs (mono-to octa-CNs) in this study were lower than those in an industrial park (183–3340 pg/g, mean 1230 pg/g) in Ningxia, China [27], while higher than those in the soil around a municipal solid waste incinerator (MSWIs) (30.1–281 pg/g, mean 87.0 pg/g) in North China [30]. The concentrations of PCNs (Tri-to octa-CNs) (73.2–295 pg/g, mean 165 pg/g) in the present study were much lower than those in an industrial park (50.0–7070 pg/g, mean 1150 pg/g) in Turkey [31], and were close to those in an industrial city (22.0–411 pg/g, mean 146 pg/g) in Pohang, South Korea [32], but higher than those in a high mountainous area (13.0–29.0 pg/g, mean 21.4 pg/g) in Sichuan, China [33].

The concentrations of ∑_18_PCBs (except CB-209) in the present study were found to be lower than those in the soil around MSWIs (28.0–264 pg/g, mean 128 pg/g) in Tianjin, China [34]. The concentrations of 7 indicator PCBs were much lower than those in the soil around a Spanish MSWI (46–5909 pg/g, mean 128 pg/g) [35]. The concentrations of dioxin-like PCBs (1.63–6.24 pg/g, mean 3.68 pg/g) in this study were lower than those in an industrial park (13.9–229 pg/g, mean 97.3 pg/g) in Shandong, China [29], and those (44–691 pg/g, mean 218 pg/g) in an industrial park in Saudi Arabia [36], while slightly lower those in a high mountainous area (7.6–10.5 pg/g, mean 9.14 pg/g) in Sichuan, China [33].

The concentrations of PCDD/Fs measured in this study were comparable to those in a high mountainous area (2.48–4.30 pg/g) in Sichuan, China [33]. Nevertheless, they were lower than those in the soil around a steel smelter (13–320 pg/g, mean 114 pg/g) in North China [37], and those in the soil around a MSWI (17.2–157 pg/g) in Harbin, China [7]. Apart from that, the PCDD/F concentrations were also lower than those (23.2–79.0 pg/g) reported by Al-Wabel [38] in an industrial area in central Saudi Arabia, and those in the soil near to a cement kiln (4.99–38.1 pg/g) in Spain [39].

In comparison, POPs concentrations in the present study, especially those of PCBs and PCDD/Fs, were lower than those in the soil around other contamination sources, such as industrial parks, steel plants, and MSWIs. This may be explained by the fact that cement production usually operate at high temperatures (>1000 °C), which prevents the incomplete combustion of solid wastes [18].

### 3.2. PCN, PCB, and PCDD/F Concentrations in Soil from Different Distance from the Stack of the Cement Kiln

In previous studies, variation trends of target pollutant levels with the distance from a certain emission source were usually analyzed to investigate the impact of the source on the surrounding environment [31,34,40,41]. In this study, 17 soil samples were divided into five groups according to the distances (0–500, 500–1000, 1000–1500, 1500–2000, and 2000–2500 m) from the cement kiln stack. The average TEQ concentrations of PCNs, PCBs, and PCDD/Fs in each group were calculated, and regression analysis was performed to verify whether the levels of PCNs, PCBs, and PCDD/Fs in soil samples decreased with the increase of distance from the stack. In addition, an exponential function equation was obtained (Figure 2). In the equation, *y* represents the TEQ concentrations of target compounds in the soil samples, while *x* stands for the distance from the cement kiln stack.

As shown in Figure 2, the highest concentrations of PCNs and PCBs were found within 0–500 m from the stack, while the highest concentrations of PCDD/Fs were found within 1000–1500 m from the stack. The concentrations of PCNs decreased significantly with increasing distance from the stack (*R*^2^ = 0.8231), and the concentrations of PCBs decreased slightly (*R*^2^ = 0.548). However, the concentrations of PCDD/Fs did not exhibit the same trend. These results suggest that the cement kiln co-processing municipal wastes have limited effects on the surrounding soil contamination with PCDD/Fs, but may exert potential effects on PCNs and PCBs. In this study, PCNs in soils were greatly influenced by cement kiln co-processing municipal wastes. In particular, high TEQ concentrations of PCNs were identified in the soil samples within 500 m from the stack, which are consistent with the previous study findings. Tian et al. observed the highest PCNs concentrations within 500 m from the MSWIs stack, and the concentration of PCNs decreased with increasing distance from the MSWIs stack [30]. Moreover, the unidentified combustion sources from residents and vehicles in the studied area might also make some contributions to concentrations of PCDD/Fs [40], which perhaps explains the high TEQ concentrations of PCDD/Fs in soil within 1000–1500 m from the stack.

### 3.3. Homologue Patterns of PCNs, PCBs, and PCDD/Fs in Soil from Different Distance from the Stack of the Cement Kiln

The profiles of PCN, PCB, and PCDD/F homologues in soil samples in five groups as mentioned above and the fly ash sample are shown in Figure 3. As shown in Figure 3a, the soil samples within 0–500 m were dominant by di-chlorinated naphthalenes (DiCN) accounting for 56.1%, followed by mono-chlorinated naphthalenes (MoCN) and tri-chlorinated naphthalenes (TrCN) with 18.3% and 20.4% contributions, respectively. Although the DiCN, MoCN, and TrCN were also the dominant homologue in the soil samples within 500–1000 m and 1000–1500 m, DiCN contributed less to the total PCNs, which was 37.8% and 36.8%, respectively. The profiles of PCN homologues varied with increasing distance, with DiCN (34.3%) and TrCN (35.2%) occupying dominant contributors in the soil samples beyond 1500 m in contrast to a small contribution of MoCN (6.4%). In addition, this study analyzed the similarity of PCN, PCB, and PCDD/F homologue profiles between each group of soil samples and fly ash samples by means of correlation analysis (Appendix A). It was found that PCN homologue profiles in soil samples within 0–1500 m were more similar to the fly ash sample (r_0–500 m_ = 0.994, r_500–1000 m_ = 0.873, r_1000–1500 m_ = 0.926, *p* < 0.01; r_1500–2000 m_ = 0.655, *p* > 0.05; r_2000–2500 m_ = 0.750, 0.01 < *p* < 0.05), indicating that the cement kiln co-processing municipal wastes could impact the profile of PCNs homologue in the soils within 0–1500 m from the cement kiln stack.

Figure 3b shows the PCB homologue profiles in soil and the fly ash sample. The contribution of low-chlorinated PCBs was significantly higher than that of high-chlorinated PCBs, and the contribution rates of tri-chlorinated to hexa-chlorinated PCBs (Tr-HxCB) ranged from 89.4% to 96.8%, with an average of 94.4%. TrCB was the primary contributor, accounting for 46.7%–59.7% of total PCBs, with an average of 51.4%. In the soil samples within 0–2000 m, tetra-chlorinated PCBs (TeCB) was the secondary contributor, with 15.3–21.0% (mean 17.8%) contribution. However, HxCB (24.6%) was the secondary contributor to total PCBs in the soil samples beyond 2000 m. Moreover, the contributions of the PCB homologues to the total PCB concentrations in soil samples from 0–2000 m decreased with the degree of chlorination increased. Correlation analysis showed that the PCB homologues were similarly distributed in soil samples within 0–2000 m and the fly ash sample (r_0–500 m_ = 0.958, r_500–1000 m_ = 0.925, r_1000–1500 m_ = 0.940, r_1500–2000 m_ = 0.976, *p* < 0.01; r_2000–2500 m_ = 0.751, *p* > 0.05). The observations above suggest that the cement kiln co-processing municipal wastes may impact the distribution of PCBs homologue in soil samples less than 2000 m from the stack.

Figure 3c displays the PCDD/F homologue profiles in soil and the fly ash samples. The HpCDF was the primary contributor in the fly ash sample, accounting for 54.9% of the total PCDD/Fs. While the PCDD/F homologue profile in each soil sample was similar, OCDD was the dominating contributor, with 32.2–48.6% (mean 41.2%) contributions. Correlation analysis suggested a low similarity of PCDD/F homologue profiles between soil samples and the fly ash sample (r_0–500 m_ = 0.046, r_500–1000 m_ = 0.127, r_1000–1500 m_ = 0.001, r_1500–2000 m_ = 0.078, r_2000–2500_ = 0.282, *p* > 0.05). In addition, many studies have suggested that when the ratio of TEQ concentration of PCDF to the PCDD is higher than 1, there are probably combustion sources of PCDD/Fs in the study area [41,42,43,44]. The PCDF/PCDD ratio of less than 1 in this study also suggests that PCDD/Fs in the soils may have been less impacted by cement kiln co-processing municipal wastes.

### 3.4. PCA of PCNs, PCBs and PCDD/Fs

Principal component analysis (PCA) was used to investigate whether the cement kiln co-processing municipal wastes was responsible for the soil contamination with PCNs, PCBs, and PCDD/Fs in the study area. The PCA score plots with the samples analyzed is shown in Figure 4. Component 1 and component 2 of PCNs accounted for 40.6% and 30.5% of the total variance, respectively (Figure 4a). Samples with high TrCN concentrations are plotted toward the negative direction of component 1, whereas samples mainly influenced by the hepta- and octa-CN are located to the right. The negative direction of component 2 featured samples that were dominated by MoCN and DiCN, and those in the positive direction were mainly affected by tetra-, penta-, and hexa-CN. Figure 4b shows the PCA score plot of PCBs, with the contribution of 63.9% and 24.3% for component 1 and component 2, respectively. Component 1 was mainly affected by deca-CB in the positive direction and by TrCB in the negative direction. Component 2 was affected by the penta-CB in the positive direction and by the HxCB in the negative direction. Figure 4c showed the PCA score of PCDD/Fs. Component 1 contributed 34.5% of the variance and was mainly affected by the PCDF in the positive direction and by PCDD in the negative direction. Component 2 accounted for 21.7%, and was mainly affected by the less-chlorinated PCDD/Fs in the positive direction. As shown in Figure 4a and Figure 4b, most of the soil samples were clustered into one group, and the fly ash sample was also located within the group. Figure 4c shows that the soil samples were clustered, but the fly ash sample was far away from the soil samples. The PCA results confirm that the cement kiln co-processing municipal wastes has potential effects on PCNs and PCBs in the soil, but limited effects on PCDD/Fs concentrations.

Mass balances are usually used to evaluate the net emissions of pollutants from thermal emission sources [19,45]. The results reported above could be attributed to the different mass balances of the PCNs, PCBs, and PCDD/Fs of the cement kiln co-processing municipal wastes. The reduction efficiency of PCBs and PCNs during cement kiln co-processing waste were 85% and 54%, respectively [3,21], while the reduction efficiency of PCDD/Fs was 94% [19]. In comparison; among these three compounds emitted from the cement kiln co-processing municipal wastes; PCNs were the primary pollutants, followed by PCBs, and PCDD/Fs were much less emitted.

### 3.5. Risk Assessment of PCNs, PCBs, and PCDD/Fs in the Soils

Risk assessment results (Appendix A) show that the carcinogenic risks (CR) for children and adults were 1.65 × 10^−8^–8.93 × 10^−8^ and 1.70 × 10^−8^–9.16 × 10^−8^, respectively, which was less than the carcinogenic risk threshold (CR = 1 × 10^−6^). The non-carcinogenic risks (no-CR) for children and adults were 1.58 × 10^−3^–8.55 × 10^−3^ and 1.57 × 10^−4^–8.48 × 10^−4^, respectively, which were much lower than the reference value (non-CR = 1). The contributions of the CR and non-CR to the total risks via three pathways (ingestion, dermal, and inhalation) are shown in Appendix A. It suggested that incidental ingestion (90%) was the main pathway of exposure, followed by dermal (7%) and inhalation (3%). Furthermore, children in the study area face greater no-CR than adults, mainly contributed by the following actors: (1) the children have more hand-to-mouth behavior to accidentally ingest pollutants [46]. (2) The children were assumed to ingest 200 mg per day while the adults ingest 100 mg per day, and the children (15 kg) have lower body weight than adults (80 kg).

## 4. Conclusions

This study analyzed the concentrations of PCNs, PCDD/Fs, and PCBs in soil samples around the first cement kiln co-processing municipal wastes in northwest China. PCNs were found to be the primary contributors to the total POP concentrations, while the PCDD/Fs dominated the TEQ concentrations. Compared with the concentrations of PCNs, PCBs, and PCDD/Fs in the soils around other thermal emission sources and industrial parks, the concentrations of PCNs, PCBs, and PCDD/Fs in this study were relatively low. The soil contamination with PCNs and PCBs may be affected by the cement kiln, but with lesser impact on PCDD/Fs. Health risk assessment results showed that the carcinogenic and non-carcinogenic risks of PCNs, PCBs, and PCDD/Fs for children and adults in soil are acceptable, and the incidental ingestion (90%) was the primary exposure pathway.

## Figures and Tables

**Figure 1 ijerph-19-12860-f001:**
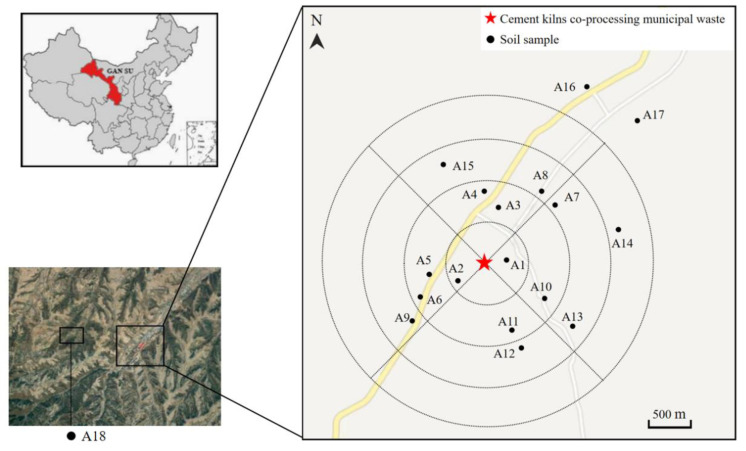
Map of the area investigated showing the soil sampling locations, where A18 denotes the background sample.

**Figure 2 ijerph-19-12860-f002:**
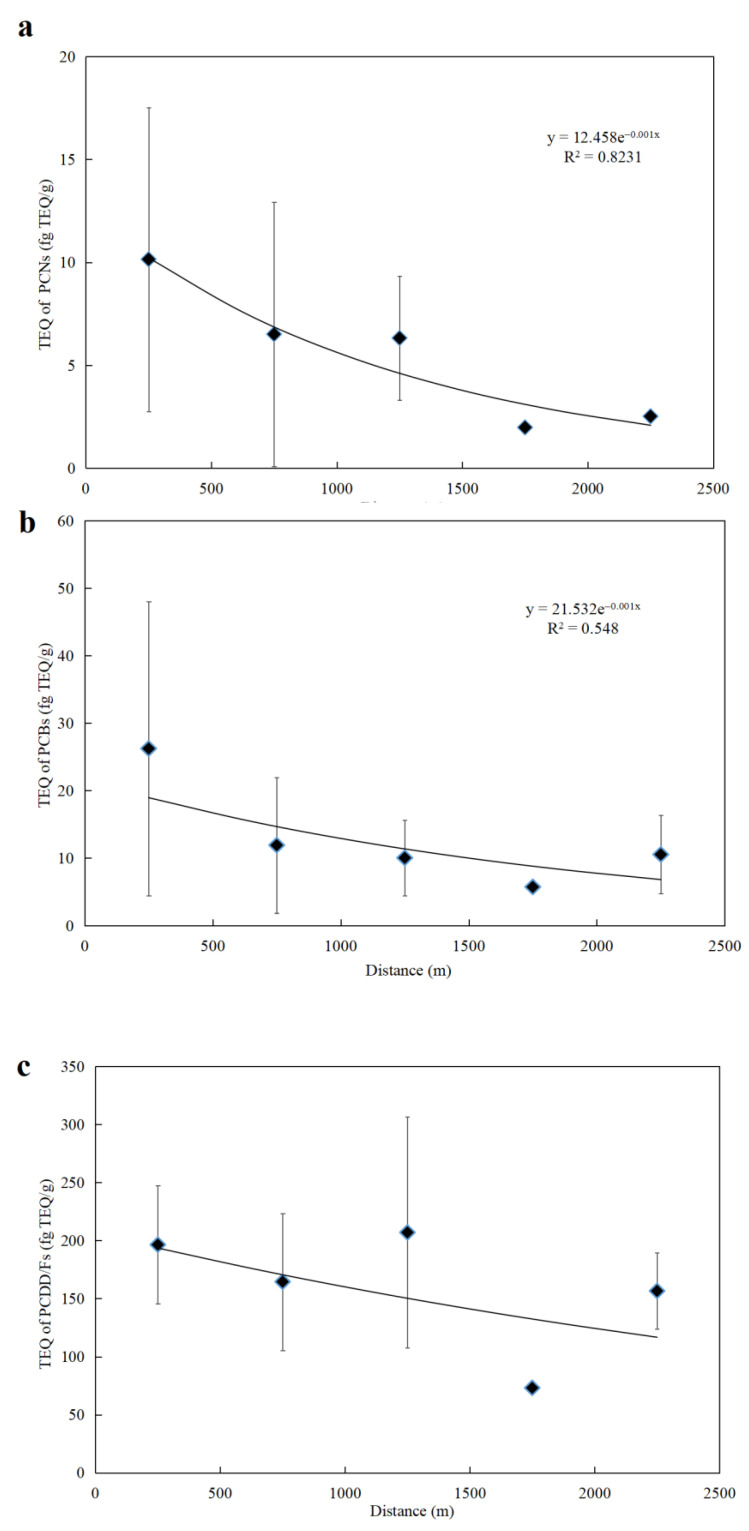
Relationship between distance from the cement kiln co-processing municipal wastes stack and TEQ of PCNs (**a**), PCBs (**b**), and PCDD/Fs (**c**).

**Figure 3 ijerph-19-12860-f003:**
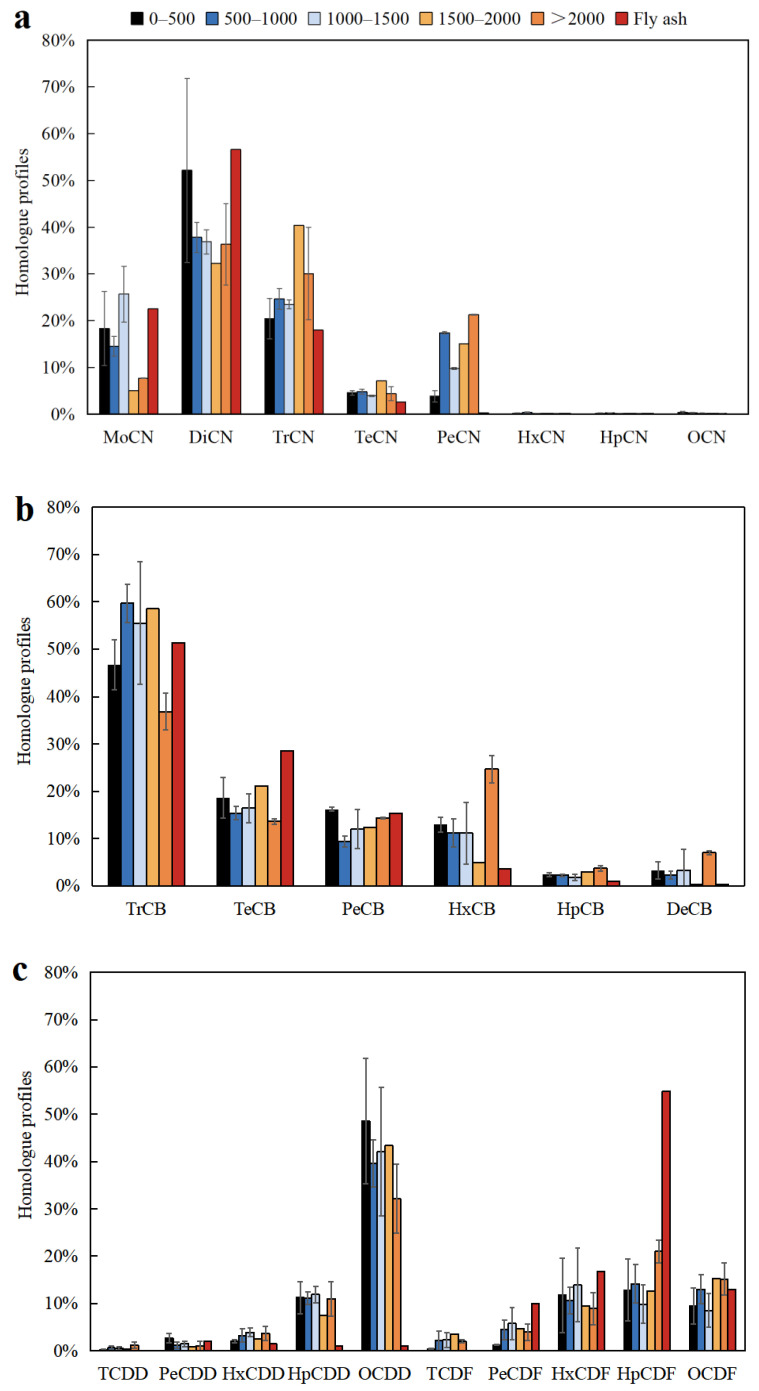
Homologue patterns of PCNs (**a**), PCBs (**b**), and PCDD/Fs (**c**) in the fly ash and soil samples.

**Figure 4 ijerph-19-12860-f004:**
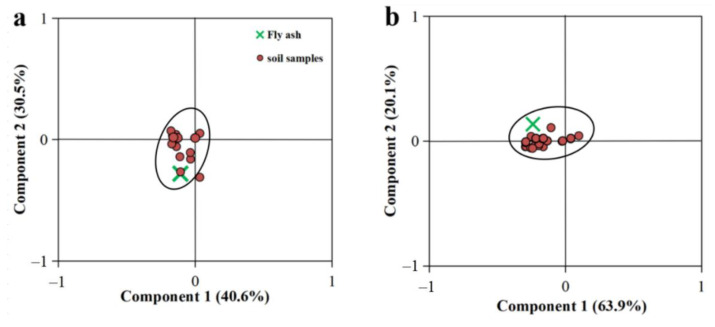
Principal component analysis score plots for PCN (**a**), PCB (**b**), and PCDD/F (**c**) homologue concentrations in the soil and fly ash samples.

**Table 1 ijerph-19-12860-t001:** Concentrations of PCDD/Fs, PCBs, and PCNs of the soil samples and background (BKG) site.

	Concentrations (pg/g)			ΣTEQ (fg WHO-TEQ/g)		
	PCNs	PCBs	2,3,7,8-PCDD/Fs	PCNs	PCBs	PCDD/Fs
A1	1288	23.0	5.95	17.5	48.0	247
A2	293	35.2	2.89	2.77	4.45	146
A3	138	20.1	4.83	3.99	31.9	141
A4	368	46.1	3.55	4.52	11.9	128
A5	277	33.1	3.31	3.55	14.5	161
A6	385	44.0	3.52	3.96	8.25	127
A7	390	32.9	3.04	4.60	2.74	141
A8	313	22.4	2.62	4.18	9.14	158
A9	383	36.0	3.64	4.53	4.72	121
A10	148	21.6	4.44	2.27	2.48	136
A11	486	40.9	4.58	20.8	2.36	294
A12	326	21.1	3.69	8.43	9.93	216
A13	1004	10.8	4.66	12.2	6.38	418
A14	299	18.1	2.50	1.98	5.73	73.2
A15	416	59.5	5.09	4.04	20.0	186
A16	132	21.2	3.92	2.66	16.3	124
A17	305	41.6	2.98	2.38	4.74	189
BKG	40.0	11.0	1.90	0.83	6.74	72.5

**Table 2 ijerph-19-12860-t002:** Soil POP concentrations (pg/g) and TEQ (pg WHO-TEQ/g) were reported in previous studies.

	Sites	Area	Concentration	TEQ	References
PCNs	Ningxia Province, China	Around an industrial area ^c^	183–3340	0.01–0.81	[27]
	North China	Around the MSWI ^e^	30.4–281	0.008–0.130	[30]
	Dilovasi, Turkey	Around an Industrial area ^d^	50–7070	0.10–1483	[31]
	Pohang city, Korea	Close to industrial area ^d^	22.0–411	0.007–0.069	[32]
	Sichuan Province, China	Tibet–Qinghai Plateau ^a,d^	13.0–29.0	0.001–0.070	[33]
	Gansu Province, China	Around the cement kiln co-processing municipal wastes ^c^	138–1288	0.002–0.021	Present study
PCBs	Tianjin city, China	Around the MSWI ^h^	28–264	0.02–0.18	[34]
	Catalonia, Spain	Around the MSWI ^f^	46–5909	/ ^b^	[35]
	Shandong Province, China	Around an Industrial area ^g^	13.9–229	/ ^b^	[28]
	Central region, Saudi Arabia	Around an Industrial area ^g^	44–691	0.34–1.97	[36]
	Sichuan Province, China	Tibet–Qinghai plateau ^g^	7.6–10.5	0.01–0.02	[33]
	Gansu Province, China	Around the cement kiln co-processing municipal wastes ^i^	10.8–59.5	0.003–0.048	Present study
PCDD/Fs	Sichuan Province, China	Tibet–Qinghai plateau	2.48–4.30	0.27–0.40	[33]
	North China	Around the iron and steel plants	13–320	0.16–4.5	[37]
	Harbin city, China	Around the MSWI	17.2–157	0.59–8.81	[7]
	Central region, Saudi Arabia	Around an Industrial area	23.2–79.0	1.73–3.98	[38]
	Catalonia, Spain	Around the cement kilns	4.99–38.1	0.19–0.80	[39]
	Gansu Province, China	Around the cement kiln co-processing municipal wastes	2.50–5.95	0.073–0.418	Present study

^a^ Remote areas without pollution sources; ^b^ Not reported; ^c^ Mono-to octa-CNs (the number of the congers was 75); ^d^ Tri-to octa-CNs (the number of the congers was 32); ^e^ Mono-to octa-CNs (the number of the congers was 24); ^f^ Tri-to hepta-CBs (the number of the congers was 7); ^g^ Tri-to-hepta-CBs (the number of the congers was 12); ^h^ Tri-to-hepta-CBs (the number of the congers was 18); ^i^ Tri-to deca-CBs (the number of the congers was 19).

## Data Availability

Not applicable.

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
