# Peer review of "PCNs, PCBs, and PCDD/Fs in Soil around a Cement Kiln Co-Processing Municipal Wastes in Northwestern China: Levels, Distribution, and Potential Human Health Risks"

_ijerph, 2022, doi:10.3390/ijerph191912860_

Round 1

Reviewer 1 Report

Authors investigated the contamination characteristics of PCNs, PCBs and PCDD/Fs in soil around the first cement kiln co-processing municipal wastes in North-west China. The presented publication is important for the local population living in the vicinity of the cement plant and for the natural environment. Such studies require detailed information about the technology of the production process. Production volume, raw materials used, their quality, technologies used to reduce dust emissions, annual emissions, monitoring of pollutants. The data on the technological process contained in the text are general and say little about the emitter of pollutants. Eg chimney - how high, actual production is about 110,000 tons per year? What is the monitoring of pollutant emissions from the chimney? The data contained in the publication shows that the production plant has been operating since 2015. In which year was the research conducted? Does the research concern 2022, i.e. after 7 years of operation of an industrial plant?  What is the main pollutant emitted from the factory (annual emissions, field distribution studies). The last information important for this type of research is information on the amount of precipitation and the average number of sunny days per year. These two parameters, apart from the wind direction, significantly determine the distribution of pollutants in the vicinity of industrial plants. In the paper, a discussion should be added on: whether the PCNs, PCBs and PCDD/Fs deposit in the soil is constant or grows every day / month / year during industrial production. Is only this one production plant located in the industrial center? What other industrial plants or housing estates may be pollutants in the studied area?

All abbreviations must be explained in the text when they are first used.

Reviewer 2 Report

Attached.

Round 2

Reviewer 1 Report

Currently, the publication contains all the data to be accepted for publication in the journal.